# Therapeutic Nature Activities: A Step Toward the Labor Market for Traumatized Refugees

**DOI:** 10.3390/ijerph17207542

**Published:** 2020-10-16

**Authors:** Dorthe Varning Poulsen, Anna María Pálsdóttir, Sasja Iza Christensen, Lotta Wilson, Sigurd Wiingaard Uldall

**Affiliations:** 1Department of Geosciences and Natural Resource Management, University of Copenhagen, 1958 Frederiksberg, Denmark; 2Department of Work Science, Business Economics and Environmental Psychology, Swedish University of Agricultural Sciences, 23053 Alnarp, Sweden; 3New Roots, Hallingelille Økosamfund, 4100 Ringsted, Denmark; sasja@siza.dk (S.I.C.); lottawilson@hotmail.com (L.W.); 4Competence Centre for Transcultural Psychiatry (CTP), Mental Health Centre Ballerup, 2750 Ballerup, Denmark; sigurd.wiingaard.uldall@regionh.dk

**Keywords:** refugees, migrants, horticultural activities, recovery, vocational program, labor market

## Abstract

*Background*: Globally, the number of refugees is growing. For many refugees, entering the labor market in their new country of residence is challenging. Some remain forever dependent on welfare services, and this not only weakens their chances of integration, but also harms their health and well-being. *Methods*: This qualitative single case study focused on a group of war-stricken refugees in Denmark. The study investigated the impact of an eight-month horticultural vocational program aimed at improving their ability to complete an education program or to work. A total of 29 interviews were conducted and analyzed using the interpretative phenomenological analysis (IPA) method. *Results*: The natural environment in the eco-village evoked a feeling of safety as well as positive memories in the participants, in contrast to the traumatic memories they had of their flight. Horticultural activities and the positive and respectful attitude from staff initiated a recovery process. New skills were achieved at an individual pace, and feelings of isolation decreased. These findings can be implicated in future interventions.

## 1. Introduction

Worldwide, the number of migrants is increasing dramatically and was estimated as 272 million in 2019 [1]. With 82 million people, Europe hosts the highest number of migrants in the world, followed by Northern America with 59 million people. In 2019, 676,300 asylum seekers applied for international protection [2]. The UN Refugee Agency [3] defines a refugee as a person who has fled an armed conflict or persecution and who is protected by international law, whereas a migrant chooses to move to a different place mainly to improve their living conditions. According to the UN Refugee Convention [4], if a migrant seeks asylum in a country and is found to be entitled to a permanent residence permit, they have the status of refugee.

After a migrant has achieved asylum, the process of integration begins. This process comprises many different steps, with securing a job being an important one. According to the European Commission, the average employment rate of working-age, non-EU migrants residing in the EU was 55% in 2017. In comparison, the figure was 68% for host-country nationals. A 2011 report by the European Commission showed that the employment rate of refugees was related to their level of education: refugees with a high level of educational attainment had a higher employment rate (70%) compared with refugees with a low level of educational attainment (45%). The employment rate for refugees with poor language skills was even lower (27%) [5]. Moreover, 39% of non-EU migrants (or 5.7 million) live in relative poverty, compared with 17% of host-country nationals [6]. Several factors can explain this economic discrepancy including language difficulties, cultural barriers, social isolation, and mental health issues [7,8,9] such as depression and post-traumatic stress disorder (PTSD) [8,10,11].

### 1.1. Nature as a Part of Restoration from Trauma

In the last decade, research has documented that nature contributes positively to our mental state in numerous ways [12,13,14]. Among other things, nature-based therapy and horticultural therapy have been shown to alleviate symptoms associated with PTSD [15,16,17,18,19,20], and to help people suffering from anxiety or depression [20,21,22]. Nature-based therapy is defined by Corazon et al. [23] as “[a]n intervention initiating a therapeutic process with activities involving natural elements in a specially designed or chosen natural environment, aiming for the recovery for a specific patient group”. Horticultural therapy has a similar goal, but the setting is restricted to a garden. Kam et al. [24] defined horticultural therapy as “[t]he use of plants as a therapeutic medium by a trained professional to achieve a clinically defined goal”.

Up until now, few studies have investigated the impact of nature-based therapy. One of these studies is a protocol paper [25] that describes a project for a group of migrants participating in a 12-week program with nature-based activities in a therapy garden that aimed to strengthen their resilience. A 2019 review by Gentin et al. [26] analyzing 11 European journal papers linking nature and integration found that nature should be understood as a resource and a means to integration, and recommends that more studies be conducted in order to understand how nature can support the health and well-being of refugees as well as their social interaction with locals. The knowledge gap prompted a recent call for studies to investigate nature’s role with regard to the mental health of refugees as well as their social interaction with locals.

### 1.2. The Situation in Denmark

Denmark has a population of 6.4 million people. Since 1997, the number of refugees or reunited family members of refugees who have been granted residence has amounted to 66,000 [27]. The overall national employment rate is 73.6%. However, the employment rate for refugees is only 33% [28]. It is estimated that due to mental health issues, 30% of refugees in Denmark are unable to enter the labor market without first receiving health treatment or completing a labor-market integration program [29,30,31]. This figure is comparable to that of other Western countries [26,27]. It is a target of the current government for at least 50% of all refugees to hold ordinary employment [32]. This target of ensuring the fast entry of refugees into the labor market is based on concern about the risk of clientalization (i.e., the situation where a citizen becomes a client who depends on welfare subsidies) [33]. However, this fast-track approach risks being counterproductive because of the lack of focus on the individual’s previous education or a failure to consider mental health problems such as PTSD [30]. The situation in Denmark is that too many refugees (according to the target set by the government) become permanently dependent on welfare subsidies, which is cost-intensive for society and does not stimulate the integration process.

### 1.3. Aim

The aim of the present study was to examine how refugees experienced an eight-month horticultural vocational program intervention with respect to their mental, social, and physical resources. An additional aim was to investigate which aspects they considered important factors in bringing them closer to education and employment.

### 1.4. The Research Questions

How did refugees outside the labor market experience an eight-month horticultural vocational program, taking into account their overall resources?

Which aspects of the program did the refugees identify as important factors for approaching education or employment?

## 2. Materials and Methods

The project GROW ran from September 2016 to June 2019 and was a collaboration between the social care farm New Roots in Zealand, Denmark, the Competence Center for Transcultural Psychiatry (CTP), and the Department of Geosciences and Natural Resource Management at the University of Copenhagen. The overall study was conducted as a three-year longitudinal single case study investigating the refugees’ perception of an eight-month horticultural vocational program. The current evaluation is part of a larger mixed-method study examining the impact of an eight-month horticultural vocational program for refugees aimed at regenerating the physical, mental, and social competences needed to retain a job or complete an education. Another component of the project concerned the effect of the intervention on job affiliation, level of functioning, and mental health. This component is reported separately.

### 2.1. The Setting

Hallingelille is an ecovillage located in the middle of Zealand [34]. It is home to 80 residents ranging in age from 0 to 75 (approx. 50 adults and 30 children). Figure 1 gives an overwiev of the village.

The majority of the residents are families who live in individual houses or in collectives. Some of the residents work outside the village, and some work from home. They are all committed to a self-sufficient ecological lifestyle, and contributing to the community is a high priority. The village is approximately 25 hectares in size including vegetable fields, a lake, a forest, and pens and stables for sheep and horses. Part of the area is specifically constructed to reduce stress based on inspiration from the design of the therapy garden Nacadia [23]. This area comprises a fire pit shelter, an outdoor kitchen, a greenhouse, and an indoor mulch toilet.

### 2.2. The Participants

Inclusion criteria were as follows: (1) refugees granted resident permit; (2) started (or completed) an education program at the Danish language school; and (3) assessed as being not ready for work due to health issues despite not having a medical diagnosis.

Exclusion criteria were suicide risk, substance abuse, or severe mental illness that required specialist treatment.

Refugees originating from seven countries participated in the GROW project (Syria, Congo, Iran, Iraq, Afghanistan, Kurdistan, Eritrea, and Chechnya, a federal subject of the Russian Federation), and the following languages were spoken: Arabic, Persian (Farsi), Chechen, Tigrinya, Kurmanji, and English.

For the whole GROW project, 55 people were screened. A total of 52 met the inclusion criteria and 37 people (22 men and 15 women) started in the project. Nine participants left the program due to severe mental illness (*n* = 2), pain that made participating not feasible (*n* = 5), or because they were to start a job or education program (*n* = 1). Altogether, 28 completed the program (17 men and 11 women). When participants, based on the screening, showed severe mental illnesses, they were advised to contact their general practitioner. Eight of the enrolled refugees were illiterate. Three of the female participants had completed a recognized education program or had worked outside the home in their homeland. The male group comprised individuals with a university degree, businessmen, soldiers, and workers (see Table 1). All 28 participants participated in individual or focus group interviews. Table 1 gives an overview of the individuals who are quoted in the analysis. The participants in the individual interviews represented different gender, age, nationality, and education level, and were selected with a view to gaining very diverse viewpoints on the data material. All names used in this study are pseudonyms.

Meetings were set up with eight municipalities located within a 50-kilometre radius of the ecovillage, and information about the project was provided. Six municipalities agreed to enroll in the project as a part of their refugee integration programs. A website was set up with pictures and a film of the activities in the ecovillage, along with written information in five languages. Neither the municipality nor the participants incurred any expense in connection with participating in the program.

The process of enrolling the participants was performed in four steps: (1) General information about the GROW project: a social worker in the respective municipality informed each refugee about the program; (2) If the refugee decided to enter the program, an interpreter gave detailed verbal and written information about the program; (3) If the refugee agreed to participate, a screening for mental health issues, substance abuse, and suicide risk was performed by a psychologist or a trained psychology student (Mini-International Neuropsychiatric Interview, and Whodas 2.0), where at any sign of severe mental or physical issues that required treatment, a general practitioner was contacted; and (4) Once the they had agreed to participate in the program and had handed in written consent, the refugees visited the ecovillage to meet with the staff and some of the residents.

### 2.3. The Program

The overall aim of the GROW program was to offer a horticultural vocational program to a group of refugees who were not ready to manage a job or an education. As a part of the program, participants prepare a meal from the crops harvested from the garden (see Figure 2, published with the consent of the participants).

Figure 3 illustrates how the program was constructed of three periods with period 1 focusing on stress-reducing activities and awareness exercise in nature settings, period 2 focusing on communication and community and experience of horticultural activities, and period 3 focusing on strengthening work competence. The program was based on meeting the needs of the individual with regard to their mental, physical, and social level of functioning. Therefore, goals based on the International Classification of Functioning, Disability, and Health model (ICF) [35] for an individual’s level of functioning—functioning and limitations at levels of body; the activities; the participation or involvement and the environmental factors—served the basis for the program.

### 2.4. Sense of Coherence as the Theoretical Basis for the Program

Antonovsky’s autogenetic model is based on a health-disease continuum between dis-ease and ease and focuses on the origins of what creates health (strengths and resources), with a focus on the development of health. According to the model, a high level of “generalized resistance resources” and “sense of coherence” (SOC) [36] can lead to better health. This assumption is embedded in the GROW program. Antonovsky defines SOC as the ability to understand the overall situation and the capacity to use the resources available [37,38]. Three elements are fundamental to the level of SOC: comprehensibility, manageability, and meaningfulness. Comprehensibility expresses the ability to perceive events as what is experienced as meaningful, understandable, and consistent. Manageability expresses the ability to balance challenges and to cope with them. Meaningfulness indicates the extent to which an individual feels that life makes sense emotionally and is sufficiently motivated to put effort into confronting problems and difficulties. The program aimed at letting the participants “grow” through an intentional use of SOC. The entire GROW program was designed to comprise a varied level of physical, mental, and social demands. The planned activities ranged from very light work to more physically demanding tasks, so as to evoke a sense of success for the participants. Experienced bodily limitations as well as motivation for an activity were the foundation for the participants’ choice of activities. Socializing with the other residents of the village, for example, meeting for coffee or preparing meals together, was organized to create a framework for social interaction [38].

The first period focused on the individual, with emphasis on stress-reducing activities in nature and building confidence in the environment and in the group. The primary objectives were to reduce stress and perform awareness exercises in a nature setting. Examples of activities were sowing seeds, taking care of the animals, or doing outdoor relaxation exercises. In the first period, groups of two to five participants met for three hours, two times a week for ten weeks.

In the second period, the initial small groups merged into groups of eight to ten participants. Focus shifted from the individual to the group, and to the challenge of being in a group and working with others. The main objectives were communication and social cohesion when involved in the horticultural activities. Examples of activities were setting up water for the plants, weeding, and repairing broken machines. These activities were conducted together with the residents. The group met for four to five hours, two times a week.

The third period represented a normal working day. The participants wore work clothes, took scheduled breaks, and ate a packed lunch. The main objectives were to strengthen the participants’ capacity to work and to determine when the individual was ready to start work, an internship, or an education program. During this period, the activities were designed as tasks that involved working together with the others, understanding instructions, and being able to maintain focus. The work lasted for five to six hours, four times a week.

### 2.5. The Staff

The staff consisted of two therapists, one of whom was a trained natural therapist and had a bachelor’s degree in health and nutrition, while the other was a trained psychotherapist specializing in dealing with trauma. Both therapists had experience working at the Danish Red Cross refugee center.

Volunteers from the village with skills in farming, wickerwork, and different kinds of handcraft also assisted the therapists, as did social workers from the municipalities and a team of interpreters.

### 2.6. Data Collection

Interviews were chosen as a suitable method to gain insight into the participants’ experience of the vocational horticultural program. The interviews were conducted at four occasions, each with differing themes:

Baseline: Memories of nature and horticulture in homeland, experiences of being in Denmark.After 15 weeks: Experiences of activities performed.After 30 weeks: Achieved competencies, future hopes and beliefs.Six months after intervention: Present position (labor market or education) reflections from the interventions.

Data consisted of face-to face interviews both individually and in groups of four to six people with the objective of gathering both sensitive and personal experiences and feelings as well as obtaining thee reflections and considerations that could be brought up in a focus group interview. All interviews were conducted by the first author, who transcribed the recorded interviews. The individual, open-ended, in-depth interviews were conducted with five participants (three men and two women) on four occasions. In total, twenty individual interviews were conducted. The focus group interviews were conducted on three occasions (see Figure 4). Each occasion included three separate groups. Seventeen individuals participated on the first occasion, and 12 and 13 individuals participated in the second and third occasions, respectively. In total, nine focus group interviews were conducted. Thus, a total of 29 interviews were conducted.

In the focus group interviews, the participants were divided into three groups on the basis of their spoken language (Arabic, Farsi, and English/Danish). Interpreters were used in all interviews. The interpreters received written and verbal information about the purpose of the interviews and the intention of obtaining detailed answers that were to be translated as literally as possible. The interviews were transcribed verbatim. Quotations used in the paper were translated from Danish to English by the researchers and then proofread by a professional English proofreader.

### 2.7. Data Analysis

Interpretative phenomenological analysis (IPA) was deemed to be a very suitable method for examining the data. The methodology has its roots in phenomenology and is concerned with how people make sense of their own significant experiences in life [39,40].

All interviews were analyzed following the stepwise procedure of the IPA method: (1) read and re-read the transcriptions; (2) evaluate for sematic content; and (3) develop emerging themes. The next step involves clustering and labelling themes as well as going back in the text and looking for new meanings. The two first authors participated in the analysis process. Fruitful discussions were conducted on the content and meaning of the text and themes in every interview, and, following these discussions, the themes were occasionally merged or re-conceptualized. Finally, the most prominent themes emerging from the data were clustered into superordinate themes and sub-themes, and were interpreted based on the research questions.

All names that appear in the results section are pseudonyms for the sake of confidentiality. All collected data from the first to the fourth interview were treated as one, and no comparison was made between the different data occasions (interviews), as this was the most logical approach unless a clear difference between the interviews was evident.

### 2.8. Ethical Considerations

The study was conducted in accordance with the ethical principles of the World Medical Association’s Declaration of Helsinki [41]. The participants were informed verbally and in writing that they could withdraw from the project or the scientific study at any time without any consequences, in accordance with the Helsinki Health Declaration [41], and the study was approved by the Danish Data Protection Agency [42]. In addition, a guarantee regarding confidentiality of information was given. All information of a personal nature was anonymized and all names and nationalities were changed in the data analysis.

## 3. Results

### 3.1. Results

The interviews comprised three superordinate themes and eight sub-themes (Table 2).

Through the analysis, these themes were used to understand the situation of the participants from the time of their flight to their settling in Denmark as well as the challenges they face and their motives and barriers for moving toward a job or education. The themes not only concerned the participants’ struggles related to the effects of fleeing from a war zone to the uncertain conditions after arriving in Denmark, but also concerned the positive aspects they experienced in connection with nature and the garden activities—is it a necessity for human value?

### 3.2. Superordinate Theme: To Not Only Survive, but Live

This superordinate theme deals with a challenge that is experienced as being almost insurmountable by many refugees. The incidents in their homeland that caused them to flee and their violently traumatic experiences during their flight did not leave them unscarred. Arriving in a new country was meant to mark the beginning of a new life, but for the refugees in this project, things did not turn out so well.

#### 3.2.1. Sub-Theme: Hopelessness Even When in a Secure Place

It is not easy for a person to overcome the traumatic events they have been through and create a “normal” life for themselves and their family. A feeling of hopelessness enveloped the participants, and they felt that life no longer made sense. It was also clear that they lacked the resources necessary to confront the challenges they faced in their new country.

This feeling of hopelessness was described by Hani: “*I had a very hard time as a refugee. But here, I’m just getting worse and worse. It’s like I’m drowning and nobody can help me*.”

Ahmamir added: “*In a war, a missile comes down, hits you and you die. Here, you die a little every day!*”

The participants used metaphors when talking about death, and sometimes used the words “death” and “die” themselves. In relation to the SOC model, the desperate feeling of slowly fading away and the cry for help from others can be seen as a very low level of coherence.

#### 3.2.2. Sub-Theme: Continuation of the Trauma

There may be no bombs or violence in Denmark, but the trauma does not always go away. The hope for a better life for their children makes them go on, even though they struggle to manage everyday life.

Selda described it as follows:

“*We have wounds that cannot be healed. In the night, I dream of the dead. In the morning, I get up, wash my face and send my children to school. We live for them. If they get better, we will get better*.”

Here, Selda expresses how she has had to shut down her own feelings in order to bring her children to a safe place.

Another point that emerged in the interviews was the concern that the participants had for their family and friends that they had to leave behind. Both the information they receive and the information they do not get worries them.

Ali expressed it as follows:

“*How can I have a happy life here when my wife and two children were bombed and my brother is in prison?*”

This feeling of the trauma they experienced seeming to continue due to their concern for family and friends was reported by several of the refugees interviewed. This can be an important barrier to having the energy or the power needed for personal growth and to establishing oneself in one’s new country.

### 3.3. Superordinate Theme: The Value of a Human Being

This superordinate theme emerged as a reaction or reflection from the participants when talking about what constituted a meaningful life. For most of them, the feeling of loss was constant, not only in terms of loss of property, but also the loss of themselves.

#### 3.3.1. Sub-Theme: The Loss of Identity

For most of the refugees, the changes that followed the transformation from being a citizen in their native country to becoming a refugee in Denmark led to a very pervasive change of their life. Some had trouble recognizing themselves as the person they were before.

Hani describes this in the following way:

“*I live in a room with four walls and I am too afraid to go out because I won’t be able to find my way home. Before, I took care of my home and my family*.”

Ahmir, who used to own a trucking company, said:

“*I am so disappointed in myself because I’m not learning the language as fast as I should, and I can’t support my family.*”

The refugees expressed that they lacked the ability to balance challenges and cope with them due to the loss of identity they experienced after arriving in a new country. All had experienced the fear of failure and of becoming a person who did not meet their expectations. Over time, such feelings may reduce the individual’s feelings of self-confidence and their capacity to manage everyday life.

#### 3.3.2. Sub-Theme: The Value of Having a Job—Is It a Necessity for Human Value?

Having a position in a family or a society by virtue of your education, job, family status (breadwinner), or something else that you are acknowledged for is part of their identity and how they see themselves. For the male refugees, because they are the head of the family and thus also financially responsible, losing their job was devastating.

Kassin described it so:

“*A human without a job has no value. They do not contribute to the progress (of society)*.”

He continued:

“*In my country, I was a successful teacher. Here, I have no value, and I would like to feel like I am worth something*.”

This quote expresses how challenging being out of work can be, even for someone with a good education. Several of the male participants were hard on themselves for not being able to find or keep a job. In contrast to the negative feelings of not having a job, we could see how happy Ahmad was when he got a job toward the end of the project:

“*I have changed 360 degrees because I got a job as a stonecutter. I feel I have a value, and every morning I am so grateful. Now I can earn money to support my family again*.”

The analysis revealed different perspectives from the female participants. For some of them, living in a country where a high percentage of women work outside the home has led to conflict between the expectations society has of them and the expectations they have of themselves. The role of being a housewife or a mother is often given a higher priority than having a job outside the home.

Zara said:

“*Well, I have my child who I take care of. I might go to school or learn something online at some point. But right now… I have my child*.”

The female participants often brought up the issue of being a full-time mother and not feeling fit enough to work as a way of justifying their hesitation to take a job, to learn Danish, or to participate in different leisure activities. Their reservations could also relate to them doubting their ability to maintain a job. Hani expressed this in the following manner:

“*I would like to work…earlier I have said ‘yes’, but I doubt I’m ready. I still have myalgia, like muscle tensions, so I don’t feel so good. I would like to, but I am not ready*.”

She expressed an ambivalence: on the one hand, she wanted to go to work, but on the other hand, she did not feel ready yet. She said that this was because she is physically disabled. However, the real reason may be that she feels pressured to take a job that she, as a housewife who has never been employed outside the home, feels insecure about.

#### 3.3.3. Sub-Theme: The Official System as an Adversary

The refugees described how their challenges understanding Danish, not only in terms of the written and spoken language, but also the culture and expectations of the authorities, put tremendous pressure on them. They had been instructed to participate in language school and to meet with their case officer at the job center, with a view to planning their future education or job opportunities. If they did not participate, their social benefits would be reduced. One example of how a misunderstanding can result in frustration was given by Damish. She went on a short trip with a friend one weekend. The ferry briefly left Danish territory and entered Swedish territory. Consequently, her social benefit was reduced as a penalty for not following the rules that dictate that you cannot leave Denmark without gaining prior consent.

“*I did not know why they cut off my money, and I could not understand what they were talking about, because I was not aware that I had done something wrong*.”

Some of the refugees described feeling unsafe when they moved about in society, and described how this affected their ability leave their apartment. Many of them had previously experienced how the police and military abused their power, and this prior experience dominated their habits.

Zara said:

“*I do not dare to go out, I’m afraid that someone will come after me or I will be shot*.”

Uhmar added:

“*In general, we are very afraid of the police, and this brings a feeling of uncertainty into our daily life. We do what can to make things work, but then anxiety takes hold…*”

This anxiety and distrust toward the official system that permeates the lives of many refugees is a factor that is often neglected by the Danish system, which is generally characterized by a high degree of trust between citizens and the authorities.

### 3.4. Superordinate Theme: Horticultural Activities as a Tool for Personal Growth and Increased Vocational Skills

In this superordinate theme, three sub-themes emerged that elucidate how horticultural activities promoted the participants’ vigor and improved their vocational skills.

#### Sub-Themes: Nature and Garden Activities As Meaningful Tools for Joy

Some of the refugees had difficulties responding to questions about their relation to, or experiences with, nature and horticulture. Their answer was typically: “*I don’t remember very much about the past*.”

As time passed, this changed, and memories slowly resurfaced. Here, Isah spoke about spring in his home country:

“*When I was a child, my father and my brothers bicycled to Kabul in the spring, when all the cherry trees were blooming. I have actually seen flowering trees here as well*.”

Being in the project and working with plants and animals helped the participants to access positive memories at the expense of traumatic events.

All the participants worked with seeds and small plants, and went on trips to the nearby forest. Most of the participants welcomed the possibility of having their own little garden, but not all showed interest in this opportunity.

Ali summed up his experience:

“*I am an accountant, and gardening is not for me. But there was another man here with the same education, and I went with him just so we could talk. As time went by, I got my own garden. A really nice garden*.”

For some of the refugees, being in a nature environment and performing green activities steered their focus (at least for a short while) toward a more positive path. That said, from time to time, some of the participants experienced an internal conflict when they were in a nature environment. Ahmamir expressed it as follows:

“*I am sitting here and looking at the nature, and it is so beautiful. My eyes see it, and my heart is delighted, but my brain will not give me permission to stop thinking*.”

This experience of a distance between the beautiful world outside and the inner feeling of disaster seems hard to deal with. However, the activities could also strengthen the individual’s self-confidence. Initially, Selda had doubts about being capable of working with the plants, but she went on to have a positive experience:

“*In the beginning I wasn’t really good at doing anything. I mostly just sat. But then I began working with the small plants, and I found out that I could actually do it. This helped me move on*.”

This lack of confidence might stem from being a woman in a society where women do not work outside the home. Under such conditions, entering the labor market in a new country can be intimidating, and the experience of being good at something might be enough to move the individual a step in the direction of employment or education.

For all the refugees, it meant a lot that they could take all the vegetables they grew home. They often reported that their families were not only happy to receive the fruits and vegetables that they knew from their home country, but they were also happy to be introduced to new crops. Growing vegetables appears to have enhanced their experience of being able to contribute to the existence and well-being of their family, and being acknowledged by others contributed to improving their self-esteem.

The participants’ narratives about their gardens also included fun and laughter, and they had friendly competitions about who could grow the biggest pumpkin or potato.

### 3.5. Superordinate Theme: Trust–Extending Relationships

The participants’ expectations of the first meeting with the staff and the villagers were dominated by suspicion. They discussed between themselves whether they were being kept under surveillance. Uhmar stated:

“*In the beginning, we thought the intelligence service had arranged it. That it would be like a prison where they would keep an eye on us. But after 2 weeks, we could feel that this was different. And good*.”

The gatherings with the staff and the residents of the village were mentioned as important for the changes that occurred to them during the project period.

Hani says:

“*I broke my leg outside our house and I was lying at home. Then they came to visit me. I really wasn’t expecting that—it meant a lot to me*.”

These concrete actions are examples of something “extra”, an achievement that no one in the official system would offer. It goes beyond the role of the professional, and is seen as a brotherly act. Being a refugee in another country where you do not speak the language and do not know the formal and informal rules of the country puts you in vulnerable position. It might be difficult to act and react emotionally balanced when you meet people in the official system. When the participants experienced acts of human kindness from the staff, they became more open toward nature and the nature activities, which ultimately led to positive development.

Establishing positive contact with the other participants was described as being very important. All the participants expressed feeling lonely and isolated.

Ali said:

“*I was in therapy and my psychologist was the only person I talked to. I was very lonely. Now I talk to many people, and I don’t need the psychologist*.”

One might expect that refugees from the same country would have a strong relationship with one another, not least if they lived in the same area of Denmark. However, most of the participants in the project simply did not have the mental resources required to contact others or arrange social events. One of the goals of the intervention was therefore to strengthen social relationships through joint activities. A comment by Ahmad at the end of the intervention gives testimony to this goal being achieved:

“*When I came here, well, I thought I would never be able to create a positive relationship with other people again. But now I have friends here*.”

#### Sub-Theme: A way to Learn Things

When the refugees started in the project, to some it was not clear what the intention was. They wanted to have a “real” job, or, for those who felt that they were unable to work, be awarded a permanent welfare subsidy without needing to have further contact with the system. It seemed strange to them to work with horticultural activities to improve their situation. Nevertheless, the experience of being trusted and acknowledged made a difference in their attitude toward the program.

Ali reflected:

“*This place has a different atmosphere. The relationships are good. You feel freer here. Not like the projects in the municipality, with all their rules and regulations. Here, we are respected*.”

Several of the participants spoke about the special atmosphere of kindness and helpfulness. Damish described the changes that took place as the program progressed:

“*In the beginning, I couldn’t be bothered to work. But after a while, we (Isah and the staff) began to plan the days, and that helped a lot. Now I go to VUC [adult education program], and things are going very well*.”

For several of the participants, learning Danish seemed to have become easier because of the different setting. Being in the village and doing activities together with the villagers forced them to remember words. In addition, the words they used were directly linked to a thing or a specific meaning in the situation. Zara expressed this difference:

“*In the language school, we learn sentences. Here, we learn to speak Danish*.”

Working in the garden and performing the program’s horticulture activities were the basic element in the path toward an education or the labor market. A goal was set for the participants to present their crops at the ecovillage’s annual harvest market. It was a positive experience for all the participants and involved pleasant interaction with local residents. Ahmamir spoke about it this way:

“*It was very surprising to us. So many people showed up and wanted to buy our products. Many people asked questions about my plants*.”

The quotations from the participants with regard to this topic revealed a sense of pride and a feeling of being shown respect by strangers.

### 3.6. Summary of Results

From the outset, the participants expressed symptoms of stress related to several different aspects of integration such as acquiring language skills and job skills. The nature-related activities generated a feeling of being competent, of being able to learn new skills, and of having improved language skills. Moreover, the program fueled a feeling of connectedness, and for most of the participants, resulted in them feeling that they were ready for a job or education.

## 4. Discussion

The findings presented above provide an opportunity to understand the possible impact of horticultural activities on moving the individual closer to education or employment.

First, this study points out that refugees are subjected to both pre- and post-migration stressors. Despite not having a psychiatric diagnosis, all participants expressed complaints indicative of mental health issues. This fact emerged in the first superordinate theme: To not only survive, but live. Even though they lived in a country with free health care, very few participants were actually receiving mental health care. This could be because of the stigma associated with seeking help for mental illness [43,44,45,46]. In the home-country cultures of the participants, language is often lacking with regard to how to refer to specific mental disorders. Due to this, one might speculate whether the participants in fact thought their mental symptoms were related to their physical health. Untreated mental health issues could explain why the participants felt too weak to manage a job and were worried that their medical doctor had overlooked or neglected a physical disease. This again can lead to mistrust of health services [47].

The experience of daily living as stressful is an example of acculturative stress that can arise from an inability to find a balance between different cultures, as defined by the Canadian psychologist J.W. Berry: “Acculturative stress is thus a reduction in the health status of individuals, and may include physical, psychological and social aspects; to qualify as acculturative stress, these changes should be related in a systematic way to known features in the acculturation process as experienced by the individual” [48]. The participants referred to a loss of identity and even described feeling that they had to fight the government authorities. These examples of acculturative stress were also found in other studies with refugees, for example, Betancourt et al. [49] and Porter et al. [50].

The significance of an increase in mental resources after participating in the program can be analyzed using Antonovsky’s model of salutogenesis (SOC) [38]. The participants in the project showed different levels of generalized resistance, defined as the individual’s ability to cope effectively with stressors. The factors in the SOC model were found to be strongly related to perceived health, especially mental health, and the relationship was revealed in study populations regardless of age, sex, ethnicity, and nationality [51,52,53]. Looking at the themes in the present study from a salutogenesis perspective, a plausible explanation can be found: For some of the participants, the loss they experienced is huge and overshadows everything in their new life. Consequently, their self-esteem is very low, as was seen in the sub-theme “Loss of identity”. Lack of manageability is expressed when life is experienced as a struggle to survive the everyday life demands from the authorities (the superordinate theme “The official system as an adversary”). Difficulties learning a new language and mental turmoil clearly indicate the mismatch between one’s resources and the experienced environments that Antonovsky [37] proposes. This could also be interpreted as a lack of a sense of belonging, and in the long term, marginalization can obstruct integration into society [54].

In their work with individuals suffering from stress in the therapy garden in Alnarp, Sweden, Palsdottir et al. described a phase of recovery where “The safe and secure frame within which the nature-based rehabilitation could originate” [55] was the first phase.

In this first phase, the initiating restorative process was established. In the current study, we found comparable reactions from the participants: Some could not remember or describe nature sceneries or memories from their homeland early in the program. This could be due to factors such as worries in general impeding another focus, it being too painful for the individual to access these memories, or the individual using all their resources to block out negative memories, with positive memories being lost as “collateral damage”. Later in the program, when feeling safer and more comfortable, participants were able to share positive memories such as the flowering cherry trees, and garden activities were experienced as meaningful and joyful. For those participants whose recovery process was slower, the beauty of nature could not be appreciated in the present moment: “My eyes see it [the beauty of nature], and my heart is delighted, but my thoughts will not give me the permission to stop thinking.” This quote could be understood as referring to a state of rumination, characterized by a repetitive focus on the causes, consequences, and symptoms of one’s distress in general. The presence of rumination can be related to symptoms of PTSD, depression, or anxiety [46,47]. Meaningfulness was, for some participants, achieved through horticultural activities, but could also have been initiated by the therapeutic relationship. Palsdottir et al. [56] described how constructing a relationship is fundamental for the participants to benefit from the project. This is cemented by other researchers in similar interventions, for example, in a review by Steel et al. [57] that investigated the model of therapeutic relationship. In this context, it is worth considering whether informality played an important role with regard to building meaningful and strong therapeutic relationships. This may give rise to considerations about whether projects organized as a cooperation between public and private/volunteer partners might be more beneficial for refugees due to the less formal role of the staff.

As proposed by Kaplan et al. [58], nature can help with the restoration of our mental and attentional capacity, especially when the mind is perceptually overloaded by thoughts. It is known that an environment that facilitates effortless attention and a quiet mind is therapeutic [59,60]. Choosing an ecovillage in the countryside as a setting for the intervention was therefore believed to be important. Based on our findings, we also proposed that there is evidence for the positive impact of a nature environment and horticultural or nature-based activity on mental health, which is in line with the findings of other studies [61,62,63,64]. Due to their poor mental health, some of the participants needed an even more sheltered environment to support their mental restoration such as a sheltered therapeutic garden. It can be stated that, for participants who were ready, the impact of nature became evident when the horticultural activities increased in the second part of the program, where motivation was found through having one’s own garden and from growing vegetables and preparing meals. Still, for some individuals, the link between these activities and getting an ordinary job could be hard to find.

Being together and sharing the experience seemed to bring meaningfulness to the participants’ lives. This is in line with the findings of Bishop and Purcell [65] who found that gardening together with others could contribute to a sense of improved health through “doing and being” and that it helped refugees to resettle in the UK. In that way meaningful work supported mental recovery and enhanced the sense of self, and created space for learning and personal growth. Post-traumatic growth (PTG) is a phenomenon that describes how some people who endure psychological struggle experience positive growth after a period of adversity. The theory of PTG was developed by Tedeschi and Calhoun [66,67] and can contribute to explaining why some of the refugees in this study were able to see the future in a very positive light, independent of their age, family relationship, and the extent of their trauma. A qualitative study with Tibetan refugees [68] concluded that factors such as social relationships, perception of oneself as a survivor, and sense of meaning and achievement developed through participating in activities were correlated to PTG. Consequently, the study recommended integration of PRG in interventions for traumatized refugees.

### Methodological Considerations and Limitations of the Study

We have strived to conduct this qualitative study in a manner that is as credible and transparently transferable as possible. However, certain parts of the methods can be critically discussed. All refugees participated in interviews, and the group as such represents a high grade of variation. Nevertheless, it was a challenge to obtain detailed responses to the questions asked. Even though all interpreters were meticulously instructed in the importance of detailed answers and a verbatim translation of spoken words from both researchers and participants, it is possible that important information was lost in translation. Moreover, the use of interpreters may have affected the participants’ sense of confidentiality. This point can refer to anxiety with regard to sharing thoughts with strangers, lack of concentration because of rumination, or coming from a culture where mental health is unpleasant to talk about. In addition, “talking about things” might be a Western way of handling things that is not natural to the participants.

The participants had different cultural backgrounds, and each individual had their own story of flight and assault. This might give more diverse results than if a group of refugees from the same country or the same conflict had been studied. It is possible that the participants may have received adverse news from their family or authorities during the program, which could have affected the interviews. As the participants were followed in the intervention for eight months, with a follow-up six months later, and because of the large sample size, the material is comprehensive enough to base conclusions on.

## 5. Conclusions

The findings in this longitudinal single case study point to the following aspects that are important for approaching education or employment:The feeling of being safe and secure in the environment.Positive contact with staff and other participants can start the recovery process.Being met with respect as a human being improves participation in program activities.Doing meaningful activities and achieving new skills becomes an important aspect as a step toward being able to retain a job or complete an education.Finally, the learning process can be stalled by feelings of loss of identity, acculturative stress, and extension of trauma. This needs to be addressed early in the intervention in order to support the participants’ recovery process.

The GROW project seemed to prepare the trainees for entering the Danish labor market and closer to participating in educational activities afterward. This study provides insight into how and why the GROW nature-based rehabilitation program can support migrants health and well-being, but more studies are needed to fully understand the mechanism at work.

## Figures and Tables

**Figure 1 ijerph-17-07542-f001:**
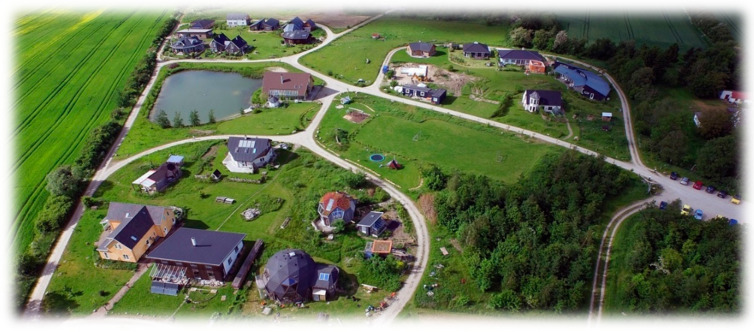
An aerial photo of Hallingelille ecovillage, showing the main housing areas and the surrounding outdoor environment of forest, lake, and agricultural fields.

**Figure 2 ijerph-17-07542-f002:**
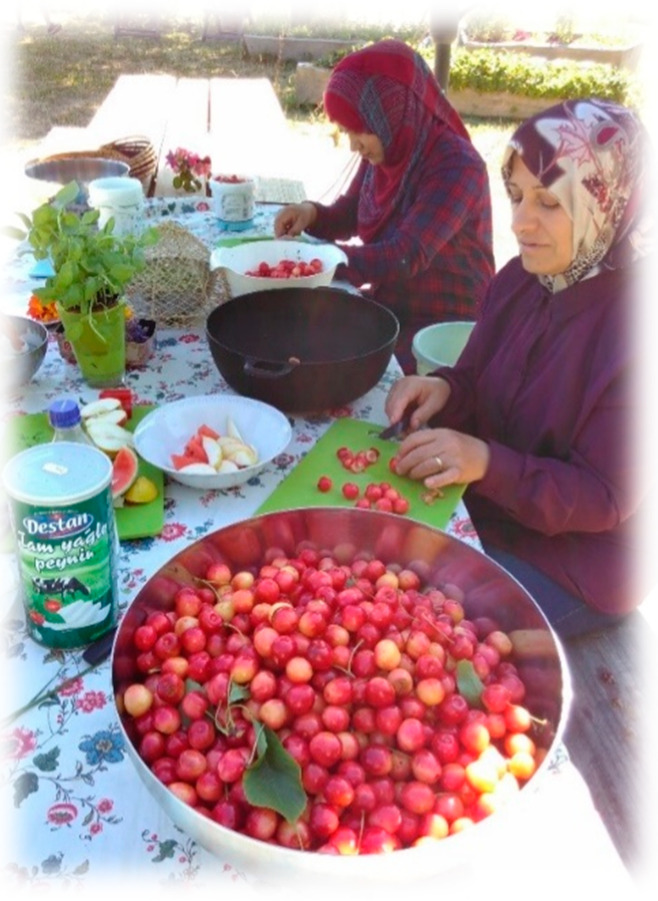
As a part of the program, participants prepare a meal from the crops harvested from the garden (published with the consent of the participants).

**Figure 3 ijerph-17-07542-f003:**
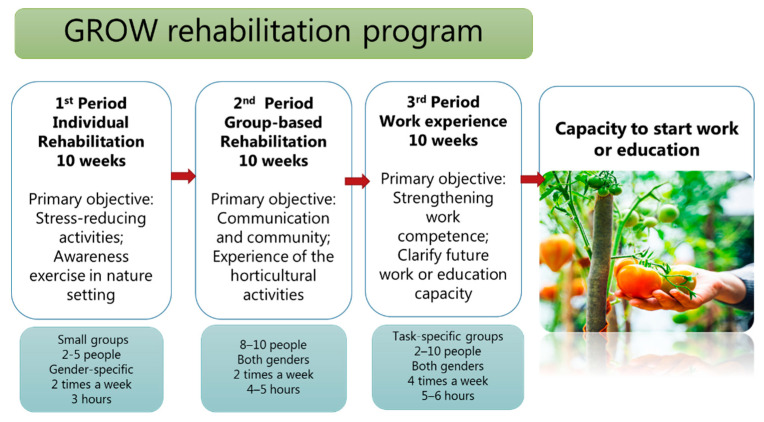
The figure illustrates the three periods in the horticultural vocational program, with period 1 focusing on stress-reducing activities and awareness exercise in nature settings, period 2 focusing on communication and community and experience of horticultural activities, and period 3 focusing on strengthening work competence.

**Figure 4 ijerph-17-07542-f004:**
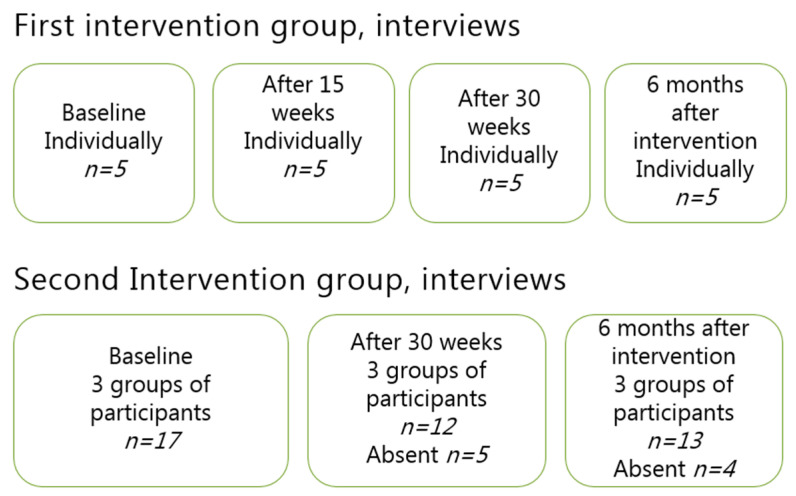
Overview of the interviews conducted in the study including both individual and group interviews. In both cases, the interviews were conducted at the start of the intervention (baseline) and on three to four follow-ups occasions (i.e., 15 weeks, 30 weeks and 6 months after the interventions ended).

**Table 1 ijerph-17-07542-t001:** Overview of the nine participants quoted in the text, indicating the pseudonyms used, gender, age, education level, or previous employment and their nationality.

Name	Gender	Age	Education/Earlier Employment	Nationality
Ali	male	63	Economist	Syria
Hani	female	52	Housewife	Syria
Selda	female	43	Housewife	Syria
Ahmamir	male	34	Driver	Palestine
Kassin	male	39	Teacher	Afghanistan
Zara	female	21	Housewife	Congo
Damish	female	28	M.Sc. in literature	Kurdistan
Isah	male	25	Artist, musician	Afghanistan
Ahmad	male	46	Blacksmith	Eritrea

**Table 2 ijerph-17-07542-t002:** Three superordinate themes emerged from the data: To not only survive, but live; The value of a human being; and Horticultural activities as a tool for personal growth and increased vocational skills. Each superordinate themes had two or three sub-themes.

Superordinate Themes	Sub-Theme	Sub-Theme	Sub-Theme
**To not only survive, but live**	Hopelessness even when in a secure place	Continuation of the trauma	
**The value of a human being**	The loss of identity	Entering the labour market—is it a necessity for human value?	The official system as an adversary
**Horticultural activities as a hool for personal growth and increased vocational skills**	Horticultural activities as meaningful tools for joy	Trust–extending relationships	A way to learn things

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
