# Peer review of "Therapeutic Nature Activities: A Step Toward the Labor Market for Traumatized Refugees"

_ijerph, 2020, doi:10.3390/ijerph17207542_

Round 1

Reviewer 1 Report

Very unique, creative, and interesting project, nicely presented. A pleasure to read. Since it is presented as a single case, the sample size, etc., are appropriate.

I found this paper presents an excellent case study that can be a model for many programs. In addition to providing refugees with usable tools that can benefit the receiving nation, it allows a population that has lost everything and been highly traumatized in the home country and in the process of fleeing, an environment that is accepting and allows them to also nurture something (the crops) from seed to harvesting to utilizing the produce.

While I am not familiar with the literature, I do know that community gardens are used effectively in many countries to help individuals living in food deserts, both by providing them healthy food as well as allowing them to develop skills, feel a sense of ownership, and have a calming influence.

Despite the relatively small sample size, this program is based on sound literature, has systematically collected data, and presented the findings cogently and compellingly.

The paper was a breath of fresh air, particularly during these turbulent times. Not only did I find it well written, I thoroughly enjoyed the outcome. Kudos to the researchers.

Author Response

Very unique, creative, and interesting project, nicely presented. A pleasure to read. Since it is presented as a single case, the sample size, etc., are appropriate.

I found this paper presents an excellent case study that can be a model for many programs. In addition to providing refugees with usable tools that can benefit the receiving nation, it allows a population that has lost everything and been highly traumatized in the home country and in the process of fleeing, an environment that is accepting and allows them to also nurture something (the crops) from seed to harvesting to utilizing the produce.

While I am not familiar with the literature, I do know that community gardens are used effectively in many countries to help individuals living in food deserts, both by providing them healthy food as well as allowing them to develop skills, feel a sense of ownership, and have a calming influence.

Despite the relatively small sample size, this program is based on sound literature, has systematically collected data, and presented the findings cogently and compellingly.
The paper was a breath of fresh air, particularly during these turbulent times. Not only did I find it well written, I thoroughly enjoyed the outcome. Kudos to the researchers.

Response: Thank you for these positive comments on our manuscript. We really appreciate your encouraging approach.

Reviewer 2 Report

The authors provide a detailed description of an interesting intervention to address the integration of refugees into their new society/nation.  The approach could be transferred to other European and North American countries.

The authors should review the manuscript for formatting and typographical errors.

In line 220, the authors state that "in-depth interviews included four participants," but in Figure 2 they note that 5 individual interviews were conducted at baseline and at each follow-up.

Author Response

The authors provide a detailed description of an interesting intervention to address the integration of refugees into their new society/nation. The approach could be transferred to other European and North American countries.

The authors should review the manuscript for formatting and typographical errors.

Response: Thank you for this response. We have addressed this matter throughout the entire manuscript. Should we have missed anything, please do not hesitate to get back to us on this matter.

In line 220, the authors state that "in-depth interviews included four participants," but in Figure 2 they note that 5 individual interviews were conducted at baseline and at each follow-up.

Response: Thank you for pointing out this discrepancy between the text and the figure. It has now been corrected in the text.  

Reviewer 3 Report

Summary:

This qualitative study evaluates the impact of an eight-month horticultural vocational program on refugee labor market integration in Denmark. Applying an interpretative phenomenological analysis (IPA) method to the total of 22 interviews, authors find that the program itself, as well as the respectful attitudes from the staff, successfully contributed to post-traumatic recovery among refugees. In particular, news skills were achieved and feelings of isolation diminished among the participants.

Comments:

  • Abstract (lines 18-19): “Methods, This qualitative single-case study focuses on this group of vulnerable refugees […]” The wording is vague here. Could the authors be more specific about which group of refugees they are referring to?
  • Abstract (line 22): “A total of 22 interviews were conducted […]” However, in lines 226-227, authors state: “In total, this yielded 20 individual interviews and 9 focus group interviews.” Can authors clarify/correct this discrepancy?
  • Lines 32-33: “[…] counting 82 million people, followed by Northern America with 59 million people.” This is wrong. According to the source authors cite here, these numbers pertain to international migrants overall, not just the refugees.
  • Lines 40-42: Since the paper focuses on refugees, it would be helpful if authors could provide the average employment rate of working-age refugees, not just the employment rate of non-EU migrants in general.
  • Line 123: Chechnya is not a country. It is a federal subject of the Russian Federation.
  • Table 1: Authors interviewed 9 participants (at four different occasions). Why these nine? Why not interview all 28 individuals who completed the program? What was the specific constraint preventing authors from interviewing everybody who completed the program? Nine is indeed a very low number for any meaningful analysis (even a qualitative one). In addition, I worry about selection, since the 9 interviewees might not be representative of the entire group of participants.
  • Figure 2: At each of the four occasions in the first intervention group interviews, the figure indicates that N=5. However, on lines 220-221, authors state that “interviews included four participants (two men and two women) at four occasions.” Why is there a discrepancy? Also, were these participants always the same four individuals? Moreover, in the second intervention group, there were 17 participants in total (although some were absent on the second and third occasion). However, Figure 1 suggests that only 9 individuals were interviewed overall. I’m very confused by this. Authors need to clearly and consistently state how many individuals they interviewed in total, and which of these individuals were interviewed just once, which were interviewed twice, and so on.
  • Line 353: There is a typo. It should read “pressured” not “pressurised”
  • Scope of the study: At the beginning, it is suggested that the study examines the impact of the horticultural program on the labor market integration of the participating refugees. However, from the results section, it is clear that very little is known about the actual impact of the program on its participants’ labor market outcomes (how likely they were to get and retain a job afterwards, what kinds of jobs they got, what was the impact on their social and economic status, etc.). Thus, authors need to be careful not to oversell their contribution. 

Author Response

Please see the attachment for our responds

Reviewer 4 Report

Thank you for submitting your manuscript “Therapeutic Nature Activities: A Step towards the Labour Market for Traumatized Refugees” to IJERPH.

The following are my comments:

  • In general, please edit for English professional proficiency.
  • Please delete extra spaces throughout the manuscript, such as in lines 53 and 57.

Abstract:

It is confusing to state that it is a single case study since it includes more than one interview. Also, please indicate that the study was performed in Denmark. 

Introduction:

  • Line 36: Please indicate the year of the UN Refugee Convention. Also, please add it to the references.
  • Line 44: Please add anxiety to the mental health issues that refugees might suffer from.
  • Nature as a part of the restoration from trauma + The situation in Denmark: please edit these sections for grammar and clarity.

Materials and Methods:

  • Please explain better how the study lasted for over 2 years when the program for refugees was conducted for only 8 months.
  • Please state that the study was conducted in Denmark and in which area.
  • Please indicate that the project was called GROW in earlier phases of the manuscript and not mention it for the first time in the participants section.
  • Line 134: Please add that the names of participants mentioned in table 1 are pseudonyms and not the real names of participants. If these are the real names of participants, please change them to protect participants. Also, are the pictures of women provided in the manuscript of real participants? Did they provide their consent to publishing them? These are serious ethical issue especially that refugee populations are usually persecuted and targeted in their home countries and can impose a real threat to their live, even after their flight.
  • Line 150: substance abuse needs to be included in the inclusion or exclusion criteria in the study and GROW project, especially that this variable was screened and tested among refugees.
  • Lines 158-159: International Classification of Functioning, Disability and Health model (ICF). The criteria of this scale needs to be introduced in the introduction.
  • All scales that were used to screen potential participants and estimate their improvement need to be detailed in measures section.
  • Sense of coherence as the theoretical basis for the programme – is this part of the scales or part of the literature review? I was confused here!!!
  • Why only 4 participants were interviewed? Who interviewed them? The same two therapists in the program? Did the participants sign any consent form? Who translated and transcribed the interviews? Were they the same interpreters who translated the participants? These issues need to be clarified.
  • Who participated in the analysis process? The authors indicate that there were discussions – with whom? Also please indicate according to what theorist the interviews were analyzed? There are many under the IPA approach.
  • What happened when participants conveyed sensitive issues or risk to self-harm or others harm during the interview or project? Were they referred to any treatment entity or professional?
  • It would have been helpful to indicate the country of origin after the quotes provided from participants and their age.
  • Line 348: the authors indicate that the following quote is from Hani [male name] and then they explain the participant’s situation as if Hani was a female – confusing!!!
  • It was not clear when they quote were coming from individual interviews or the group interviews!

Discussion:

  • The authors might want to consider adding to their discussion the term posttraumatic growth in addition to resilience they mention.
  • Conclusions need to be organized in a flowing writing, not in points for power point presentation.

Author Response

Please see the attachment for our responds. 
